# Gut Mycobiome and Asthma

**DOI:** 10.3390/jof10030192

**Published:** 2024-03-01

**Authors:** Amjad N. Kanj, Joseph H. Skalski

**Affiliations:** Division of Pulmonary and Critical Care Medicine, Mayo Clinic, 200 1st Street SW, Rochester, MN 55905, USA; kanj.amjad@mayo.edu

**Keywords:** gut–lung axis, asthma, gut mycobiome, fungal dysbiosis, *Candida*, immune pathways

## Abstract

This review explores the ‘gut–lung axis’ in asthma with a focus on commensal fungal organisms. We explore how changes to the intestinal commensal fungal community composition alter lung immune function. We comprehensively review available studies that have profiled the composition of the gut mycobiome in adults and children with asthma, and discuss mechanisms of gut–lung interactions that have been described in animal models of asthma. Studies indicate that intestinal fungal dysbiosis, such as an increased abundance of certain fungi like *Candida*, can elevate the risk of asthma in children and exacerbate it in adults. This effect is mediated through various pathways: the host immune system’s sensing of dysbiosis via C-type lectin receptors (e.g., Dectin-2), the impact of pro-inflammatory fungal metabolites (e.g., 12,13-diHOME, prostaglandin E2), and the role of lung immune cells (e.g., group 2 innate lymphoid cells [ILC2], M2 macrophages). We also describe strategies for modulating the gut mycobiome as potential therapies for severe asthma. The review concludes by emphasizing the necessity for further research into the role of the gut mycobiome in asthma to deepen our understanding of these complex interactions.

## 1. Introduction to the Gut Mycobiome in Asthma

Asthma, a widely prevalent chronic lung disorder, is characterized by airway inflammation, hyper-responsiveness, and mucus hypersecretion, leading to variable airflow obstruction [1]. Asthmatic inflammation is heterogeneous, broadly classified into type 2 and non-type-2 inflammation. Type 2 inflammation may be allergic or non-allergic, involving eosinophils, Th2 cells, group 2 innate lymphoid cells (ILC2s), and the cytokines IL-4, IL-5, and IL-13. Conversely, non-type-2 inflammation involves varied pathways including activation of Th1 and Th17 cells [2].

The gut–lung axis is the concept that changes to the composition of the intestinal microbiota can influence immune function in the lungs. This generally occurs without the translocation of microorganisms from the gut to the lung. [3,4]. Recent studies have highlighted the role of the gut–lung axis in the pathogenesis of asthma [5].

Research on intestinal dysbiosis in asthma has mostly focused on bacteria and highlighted the important role of bacterial metabolites, such as short chain fatty acids (SCFAs), in modulating allergic airway inflammation [6,7]. However, the healthy gut also contains commensal fungi, some of which, like *Aspergillus* and *Penicillium*, are known asthma triggers when inhaled [8,9]. Fungi are larger than bacteria and produce a wide range of bioactive metabolites, some with inflammatory properties [10]. Similar to bacterial communities, profiling the gastrointestinal fungal community can be performed using non-culture-based metagenomic sequencing techniques and is typically conducted through sequencing of the internal transcribed spacer (ITS) region or the 18S rRNA gene [11].

This article reviews the gut fungal microbiome (mycobiome) in asthma patients, explores important mechanisms linking the gut mycobiome and asthma from cellular and animal studies, describes strategies for modulating the gut mycobiome as therapeutic interventions in asthma, and discusses future directions in research on the gut mycobiome and asthma.

## 2. Characterizing the Gut Mycobiome in Asthma Patients

The maturation of human intestinal microbiota within the first 1 to 3 years of life has heightened research interest in the association between early-life gut mycobiome community composition and childhood asthma [12,13,14]. It is hypothesized that there may be a critical window during early development where adverse changes to the gut microbiota, referred to as microbial ‘dysbiosis’, may cause immune dysregulation that predisposes one to the onset of asthma and other allergic diseases.

A study categorizing participants by the composition of their neonatal intestinal bacterial community identified three groups with markedly different risks for atopy at age 2 years and asthma at age 4 years [15]. The highest-risk group exhibited an increased relative abundance of specific fungal genera, including *Candida*, *Rhodotorula*, *Debaryomyces*, *Meyerozyma*, *Nigrospora*, *Saccharomyces*, *Pyrenochaetopsis*, and *Phanerochaete*, and a decrease in *Malassezia*. Additionally, this group showed enrichment with various intestinal metabolites, notably the pro-inflammatory metabolite 12,13-dihydroxy-9Z-octadecenoic acid (12,13-diHOME) [15]. Enzymes producing 12,13-diHOME from linoleic acid are encoded by both gut bacteria and fungi, and pretreatment of human dendritic cells with 12,13-diHOME led to a decrease in the prevalence of regulatory T cells (Treg), a feature observed in children with uncontrolled asthma [15,16].

Another study analyzing stool samples from 3-month-old infants in rural Ecuador found an increased relative abundance of fungi in those who developed an atopic wheeze at the age of 5 years [17]. *Pichia kudriavzevii*, the teleomorph of *Candida krusei*, was significantly more prevalent in the atopic wheeze group [17]. In this study, which evaluated both bacterial and fungal communities, fungal dysbiosis was more strongly associated with later onset of atopic wheeze than observed bacterial dysbiosis. In a related study of neonatal mice with house dust mite (HDM)-induced allergic airway disease, the expansion of *P. kudriavzevii* in the gut exacerbated their condition later in life, as evidenced by heightened Th2 and Th17 cellular inflammation [18]. Interestingly, the growth and adherence of *P. kudriavzevii* to gut epithelial cells were influenced by SCFAs, highlighting the significant role of fungal interactions with bacterial-derived metabolites in the neonatal gut and their influence on asthma outcomes later in life [18]. Additionally, the interplay between bacterial and fungal populations in the gut of children was emphasized in a study by Depner et al., which demonstrated that a higher relative abundance of the prevalent fungal genus *Alternaria* at 2 months was associated with subsequent gut bacterial maturation, contributing to protection against asthma later in life [19].

*Candida*, *Saccharomyces*, and *Cladosporium* are the predominant constituents of the healthy adult gut mycobiome, with an increase in *Candida* spp. linked to diseases associated with urbanization and inversely related to bacterial microbiome diversity [20]. Less is known about the intestinal fungal mycobiome in adults with asthma. Fungal genera commonly encountered in the stools of adult asthma patients include *Candida*, *Malassezia*, *Saccharomyces* and *Aspergillus* [21,22]. In a pilot study involving 24 patients with a median age of 57 years, we demonstrated that individuals who experienced any severe asthma exacerbation (defined as an asthma-related emergency department visit or hospitalization) in the past year had a higher abundance of *Candida* spp. relative to bacteria in their gut. This finding was observed to be independent of systemic antibiotic and glucocorticoid use [22]. A larger study involving 59 participants, with a mean age of 43 years, revealed that asthma patients receiving inhaled corticosteroids (ICS) (*n* = 26) exhibited lower intestinal fungal diversity, as measured by the Shannon diversity index, compared to non-ICS recipients (*n* = 12) and healthy controls (*n* = 21) [21]. The study also identified an enrichment of the fungal genera *Russula*, *Sebacina*, *Nectria*, and *Wallemia* in the guts of asthma patients on ICS. Notably, *Wallemia*, a xerophilic spoilage fungus, has been previously shown to exacerbate asthma severity when expanded in the gut of mice with HDM-induced asthma [21,23]. Table 1 summarizes important findings from research on the gut mycobiome in patients with asthma. Importantly, it underscores the limited number of such studies presently available, particularly among adults with asthma.

## 3. Mechanisms Linking the Gut Mycobiome and Asthma

Developing animal models to investigate the role of intestinal fungal dysbiosis in asthma requires a tailored approach, dependent on the objectives of the experiment. Specific Pathogen-Free (SPF) mice are frequently used for this purpose [23]. These mice are free from specified pathogens but not all microorganisms. To promote intestinal fungal colonization, SPF mice are first treated with antibiotics to deplete gut bacterial populations, followed by fungal gavage to establish intestinal fungal colonization [23,25]. In a study investigating the effects of *Candida albicans* gut dysbiosis on colitis, antibiotics were not utilized. As a result, high doses of *C. albicans* had to be repeatedly administered via gavage to ensure effective colonization [26]. In experiments where mice were premedicated with antibiotics, a single gavage of 10^7^ live *C. albicans* yeast was sufficient to establish intestinal colonization that persisted for weeks after gavage [22].

In fact, antibiotic treatment to deplete intestinal bacteria in SPF mice is an affordable and widely accessible method. Cefoperazone, a broad-spectrum third-generation cephalosporin, is often selected for its minimal systemic absorption, making it ideal for experiments on asthma where altering the lung bacterial composition is undesirable [22,23,27]. Cefoperazone sodium salt may be dissolved at a concentration of 0.5 mg/mL in deionized water and provided to mice as the only source of drinking water for 7 days. Other antibiotics that have been utilized, often in combination, include ampicillin (1 mg/mL), clindamycin (0.5 mg/mL), metronidazole (1 mg/mL), streptomycin (5 mg/mL), ciprofloxacin (1 mg/mL), and vancomycin (1 mg/mL) [28,29,30].

Oral gavage of antibiotics has been employed in murine experiments to ensure the precise administration of antibiotic doses [31]. Although daily gavage can be labor-intensive, it may minimize variations in antibiotic intake, both within and among experimental groups, enhancing the likelihood of achieving consistent intestinal colonization. In cases where fungal colonization remains challenging, antifungals may be employed selectively to manage competing fungal species. The antifungal fluconazole has been used in studies aiming to promote the growth of fungi other than *Candida* [32]. In a study of the effect of intestinal *C. albicans* proliferation on bleomycin-induced pulmonary fibrosis in mice, fluconazole (0.5 mg/mL) was administered to the control group to ensure that no endogenous *Candida* would remain and proliferate in the gut following bacterial depletion by antibiotics [29]. A separate set of experiments was necessary to exclude the possibility that fluconazole itself influenced pulmonary fibrosis [29]. Amphotericin-B at a concentration of 0.1 mg/mL, has been shown to significantly disrupt fungal populations within the gut. Unlike fluconazole, amphotericin-B is not absorbed through the gastrointestinal tract [32].

Germ-free mice, maintained in germ-free conditions, offer a valuable alternative to SPF mice for studies requiring colonization with specific, defined organisms [30]. These mice are completely devoid of all microorganisms, enabling researchers to precisely control the microbial environment and study the effects of individual intestinal fungi or defined intestinal fungal communities on host physiology and disease. However, these mice can be costly, require specialized equipment and training, and their lack of a native microbiome may limit the generalizability of findings to microbiota-harboring organisms. This limitation is particularly relevant when considering the complex interactions within microbial ecosystems outside the gut, which are lacking in germ-free models [30].

After achieving gut dysbiosis, sensitizing the mice with airway allergens such as chicken ovalbumin (OVA) or HDM allows for the examination of the effects of intestinal fungal dysbiosis on allergic airway inflammation and its underlying mechanisms [23,33]. Airway inflammation and hyper-responsiveness can be assessed by measuring eosinophils, other inflammatory cells (e.g., Th2, ILC2), and type 2 cytokines (e.g., IL-5, IL-13) in the lungs and serum; measuring total and allergen-specific IgE in the serum; measuring airway resistance in response to methacholine; and conducting histological examinations of mucous-producing goblet cells in the airway epithelium [23,34]. Figure 1 illustrates the fundamental elements of one of several murine models of gut fungal dysbiosis and allergic airway disease. Essential to these models is confirming intestinal fungal colonization using nucleic acid amplification tests, ensuring the absence of significant gut mucosal inflammation, and verifying that the fungi administered through gavage are not present in the lungs [23,32].

Recent studies using these models have begun to unravel key pathways in gut–lung communication, enhancing our understanding of the gut–lung axis in asthma and revealing new therapeutic targets. For instance, mice with intestinal *Candida parapsilosis* overgrowth showed enhanced allergic airways disease. This effect is mediated by *Candida*-produced prostaglandins, specifically PGE2, absorbed from the gut into the systemic circulation, promoting M2 macrophage polarization in the lung and amplifying airway inflammation [28]. Cyclooxygenase inhibitors, like aspirin and celecoxib, suppress PGE2, therefore reducing M2 macrophage polarization and diminishing airway inflammation in these mice [28].

Building on these findings, Wheeler et al. administered fluconazole, targeting *Candida* spp., to HDM-sensitized mice [32]. Contrary to expectations, this led to a worsening of allergic airway inflammation. Examination of these mice’s intestinal mycobiota post-fluconazole revealed decreased *Candida* spp., but a concomitant increase in other fungi, namely, *Aspergillus amstelodami*, *Epicoccum nigrum*, and *Wallemia mellicola*. Subsequent single oral gavage of 5 × 10^6^ live conidia from these three fungal species, or from *W. mellicola* alone, into mice with allergic airway disease exacerbated their condition [23,32].

Further exploration of the *Wallemia* intestinal dysbiosis model revealed that Dectin-2, a Syk-coupled pattern recognition receptor, is pivotal for the gut–lung axis interactions aggravating asthma [27,35]. Dectin-2 recognizes high mannose structures and is a critical receptor for Th17 responses to fungal infections [36]. Dectin-2 is also expressed on both gut immune and epithelial cells [27]. Li et al. further investigated the role of Syk-coupled pattern recognition receptors in the exacerbation of asthma. They demonstrated that fungal dysbiosis aggravates allergic airway disease in mice through interactions with gut-resident CX3CR1+ mononuclear phagocytes [37]. These mononuclear phagocytes are equipped with C-type lectin receptors that trigger antifungal responses in a Syk-dependent manner [38]. Inhibiting the Syk-mediated fungal sensing in these intestinal CX3CR1+ mononuclear phagocytes was found to alleviate allergic airway disease [37].

Another potential communication pathway between gut fungi and the lung involves fungal interactions with commensal bacteria in the gut. Fungi interact with bacteria by direct physical contact, through the secretion of chemical molecules, and by utilizing metabolic by-products and competing for nutrients [39]. As previously mentioned, *Alternaria* can enhance gut bacterial maturation, contributing to protection against childhood asthma [19]. Similarly, bacteria can influence fungal maturation and dysbiosis. For example, *Enterococcus faecalis* prevents fungal overgrowth and balances the microbiome early in life [24]. This has been linked to a reduction in Th17 stimulation and a subsequent decrease in the risk of allergic diseases [40]. *Lactobacillus reuteri* is another bacterium that attenuates asthma in mice, doing so through the induction of Treg cells [41]. Interestingly, *L. reuteri* DSM 17938, commonly used in probiotic products, is well known for its antifungal effects, particularly against *P. kudriavzevii* and other *Candida* species that are in turn associated with airway inflammation when expanded in the gut [17,42,43].

Fungal dysbiosis has also been shown to affect gut barrier permeability, potentially leading to the leakage of immune cells, fungal-related products, and allergens [44,45]. Previous studies have indicated that adults with asthma exhibit increased gut permeability [46,47]. However, whether this increased permeability is a cause or a consequence of asthma remains unclear. A recent study by Schei et al., which associated early-life fungal abundance with allergic airway disease, found that increased gut permeability does not seem to be involved in the underlying mechanisms of this association [24]. Gut permeability in this study was assessed by measuring permeability markers such as lipopolysaccharide-binding protein and fatty-acid-binding protein [24]. Figure 2 describes key mechanisms by which fungal dysbiosis may exert its effects in the gut.

In the lungs, numerous changes have been associated with gut dysbiosis, including the stimulation of Th2 and Th17 pathways and the suppression of Treg cells. Notably, SCFAs, which are crucial bacterial metabolites produced by bacteria like *Lactobacillus*, play a vital role in enhancing Treg cells and, therefore, have a protective effect [6,48]. The concentration of these SCFAs in the gut may be affected by the composition of the intestinal fungal community, possibly through the interaction of fungi with commensal bacteria [49]. Recently, another cell type, ILC2, has been implicated. These cells make up less than 1% of the total lung cells but can release a substantial amount of type 2 cytokines in response to inhaled fungal allergens [50]. In a mouse model of asthma with *C. albicans*-induced gut dysbiosis, an increase in lung-resident ILC2 cells was observed even in the resting state, prior to allergen sensitization [22]. It is conceivable that these ILC2 cells become activated by epithelial cytokines, specifically IL-25, IL-33, and thymic stromal lymphopoietin (TSLP), following sensitization by an allergen. This activation could mediate gut–lung crosstalk in asthma at the level of the lungs [22,50].

## 4. Modulating the Gut Mycobiome in Asthma

Investigating the role of the gut mycobiome in asthma has uncovered potential pathways for new treatments, specifically through altering the intestinal fungal composition. This modulation of the gut mycobiota can be achieved through a variety of methods, including the use of antimicrobials and probiotics. However, research into these approaches for treating patients with asthma remains limited, with most studies conducted on mice or focused on diseases other than asthma.

The effectiveness of antifungals in treating severe asthma, particularly among patients with fungal airway sensitization and allergic bronchopulmonary mycosis, is well-supported [51,52,53]. Yet, the administration of fluconazole to mice with the aim of depleting intestinal *Candida* promoted the overgrowth of other fungal communities, which resulted in exacerbation of their allergic airway inflammation [32,54]. In a different context, fluconazole administration in mice with ulcerative colitis and intestinal *Candida* overgrowth showed an improvement in colitis [55].

Like antifungals, antibiotic treatments can disrupt the balance of gut commensal bacteria, leading to fungal overgrowth. Exposure to intrapartum antibiotics has been associated with significant changes in the gut mycobiome composition. At 18 months, infants exposed to intrapartum antibiotics exhibited a mycobiome dominated by *Trichosporon*, a commensal yeast of the gut linked to rare cases of allergic pneumonitis and invasive infections in the immunocompromised [56,57,58]. In a separate study involving 37 antibiotic-naïve infants with a respiratory viral illness, the administration of amoxicillin, with or without a macrolide, increased the relative abundance of *Candida* in the gut [59]. In fact, the decrease in the incidence of pediatric asthma in recent years may be an unintended consequence of prudent antibiotic use during infancy, which contributes to the preservation of the gut microbial community [60]. In adults, the impact of β-lactam antibiotics on the mycobiome varies between individuals, with increases in *Candida* abundance largely dependent on the individual’s existing microbiota [61].

In contrast to antimicrobials that broadly affect the microbiota and allow for the expansion of undesired commensals and pathogens in the gut, probiotics offer a more targeted approach, potentially enhancing specific beneficial microorganisms without significantly disrupting other healthy commensal populations. This strategy has particularly garnered attention in the pediatric asthma population, where the administration of bacterial probiotics has been mostly explored [62]. For instance, a randomized, placebo-controlled study involving a large cohort of schoolchildren in a primary care setting demonstrated that a probiotic mixture containing *Bifidobacterium breve* B632 and *Lactobacillus salivarius* LS01 reduced the incidence of asthma exacerbations [63]. Despite these findings, meta-analyses of randomized controlled trials have not found significant evidence to support the use of probiotic supplementation over placebo in reducing asthma risk in children [64,65]. In adults with asthma, the administration of *Bifidobacterium lactis* Probio-M8 to 17 patients was shown to decrease the level of fractional exhaled nitric oxide by day 30 and improve asthma control test scores compared to 14 patients who received conventional therapy alone [66]. Additionally, a randomized, double-blind, placebo-controlled trial involving an 8-week treatment with probiotic supplementation (Lactocare^®^) in Iranian adults with asthma led to a decrease in plasma IL-4 and improvements in forced expiratory volume in one second (FEV_1_) and forced vital capacity (FVC). The trial suggested this effect is exerted through probiotics altering the expression of microRNAs (miRNA), such as a decrease in the expression of miR-16 and miR-146a and an increase in miR-133b [67].

Fewer studies have focused on probiotics containing fungal species. One study conducted at the Universidade Federal de Minas Gerais (UFMG) that evaluated *Saccharomyces cerevisiae* UFMG A-905 demonstrated its potential in preventing the development of asthma in mice [68]. In this research, mice were challenged intranasally with OVA for three consecutive days. The administration of the probiotic was carried out daily through gavage, beginning 10 days prior to the first OVA sensitization and continuing throughout the sensitization and challenge protocol, for a total of 28 days [68]. Measures of airway hyper-responsiveness, assessed by a methacholine challenge, along with total cell and eosinophil counts, and Th2 cytokines in the bronchoalveolar lavage, were all reduced following oral administration of *S. cerevisiae* UFMG A-905. This effect was observed with a gavage of 10^9^ CFU/mL of *S. cerevisiae*, but not with lower doses [69]. This aligns with observations from other mouse experiments that demonstrate the need to deplete the microbiota for the expansion of fungal organisms, suggesting that a therapeutic strategy involving antimicrobials combined with fungal-based probiotics or even fecal microbiota transplantation to modulate the gut mycobiome in asthma may be worth exploring [70,71].

Fungal components, such as β-glucans and mannans, present intriguing potential for modulating the gut mycobiome in asthma patients. β-glucans, natural components of the cell walls of all fungi, possess well-recognized anti-inflammatory and immunomodulating properties, and have been identified as potential sources for maintaining gut microbiota and supporting the immune system [72]. In a study in children with asthma, the subcutaneous injection of β-1,3-glucan successfully reduced asthmatic symptoms and increased serum IL-10 levels [73]. Similarly, the oral administration of β-1,3/1,6-glucans (Wellmune WGP^®^) attenuated the Th2 response in a murine model of OVA-induced asthma [74]. In a randomized, placebo-controlled trial, a 4-week daily supplementation of Wellmune WGP^®^ at a dose of 250 mg/day during the allergy season to individuals suffering from ragweed allergy significantly reduced total allergy symptoms and symptom severity by 28% and 52%, respectively, and improved quality of life by 56%, but did not affect serum IgE levels [75]. On the other hand, the intratracheal administration of β-1,3-glucan to mice exacerbated allergic asthma independently of fungal sensitization and promoted glucocorticoid-resistant Th2 and Th17 responses [76]. Mannan is another fungal component and a major cell wall constituent of *S. cerevisiae*, also with potential to modulate the gut microbiota and promote airway epithelial cell repair [77,78]. Mannans were recently utilized to coat allergens, facilitating their safe and efficient uptake by immune cells for oral allergy immunotherapy [79,80]. However, their application in reducing the pathogenesis of *Aspergillus fumigatus*-induced asthma when administered intranasally has not been successful [81].

## 5. Future Directions in Gut Mycobiome and Asthma Research

The exploration of the gut mycobiome and its relationship with asthma presents a rich and relatively untapped avenue for scientific investigation. As research in this area progresses, several key directions are likely to shape the field and offer potential breakthroughs in understanding and treating asthma.

While research on the gut mycobiome in asthma has focused on early childhood, its role in adult asthma is also important. This line of investigation could uncover how microbial changes later in life impact asthma and its treatment, thereby broadening our understanding beyond the early life “critical window” [13,82]. Despite evidence suggesting that gut microbiota colonization in early life shapes the immune system, research on adults with asthma remains limited [83,84]. Nonetheless, available studies indicate that adults with asthma exhibit distinct intestinal fungal profiles. Specifically, the overgrowth of certain gut fungi like *Candida* or *Wallemia* has been associated with more severe asthma phenotypes [21,22].

To further understand the dynamics between the gut mycobiome and asthma, longitudinal studies are essential. By tracking individuals over various life stages, these studies can provide insights into how changes in the mycobiome over time correlate with asthma development, progression, and exacerbations. They can also help identify critical periods where interventions might be most effective. Expanding the research to include a wider range of ethnic and geographical populations is also important, as the gut mycobiome shows significant variability across populations, even within the same regions of the world [85]. Additionally, integrating multi-omics approaches, including proteomics, metabolomics, and transcriptomics from gut fungi, provides a more nuanced view of how alterations in the gut mycobiome may exacerbate airway inflammation [86]. Expanding this approach to include bacterial and fungal microbiome data from various body sites, particularly the lungs and upper airways, can enhance our understanding of the complex interactions and implications of microbiome changes on respiratory health [87]. In fact, fungal dysbiosis is increasingly recognized in the lungs of patients with asthma. Children with severe asthma were found to have a higher prevalence of *Pneumocystis* in their airways, linked to heightened type 2 inflammation and increased mucus production [88,89]. A different study found elevated levels of *Pneumocystis*-specific immunoglobulins G and E in the serum of individuals with severe asthma, compared to controls [90]. Moreover, adult asthma patients frequently show increased levels of allergenic molds, such as *Aspergillus*, *Alternaria*, *Cladosporium*, and *Malassezia*, in their sputum and airways [53,91,92,93]. These findings underscore the need for more extensive research into the interplay between gut and lung mycobiota in asthma’s evolution from early life to adulthood. Such a comprehensive approach allows for a more thorough investigation into the multifaceted relationship between the gut mycobiome and respiratory diseases [94].

In addition to bacteria and fungi, the human gut hosts viruses, archaea, and parasites. The interactions between these organisms and fungi in the gut remain unexplored, with very few studies investigating the role of these organisms in asthma. One study of the fecal virome in 647 infants found a significant association between several *Caudovirales* families and increased risk of asthma later in life, independent of bacteria [95]. This association was guided by a host’s toll-like receptor-9 gene variant, highlighting interactions between phages and the host immune system [95]. Another study using fecal samples from 472 children aged 6 to 10 years showed that the archaea *Methanosphaera stadtmanae*, detected in 78% of samples, was associated with a lower risk of asthma during this age range [96]. Regarding helminths, or parasitic worms, their relationship with asthma and allergies is nuanced, with both the potential to exacerbate and mitigate these conditions [97]. Research highlights how intestinal helminths can modulate immunity and inflammation through the induction of Treg cells and regulatory macrophages [97]. Specifically, a study on *Ascaris lumbricoides*, *Trichuris trichiura*, and *Toxocara* spp. suggested that helminthiasis may enhance Th2 immune responses and foster an immune regulatory phenotype that suppresses allergic reactions, although no direct link to asthma was identified [98]. Conversely, a case-control study found that *Trichuris trichiura*, but not other helminths, in the stools of children aged 6–14 years is associated with asthma, with detection of helminths achieved through flotation and microscopic analysis [99]. Although no research has directly examined the interactions between parasites and gut mycobiota in patients with asthma, investigations in non-human primates suggest potential interactions, with *Strongyloides* negatively correlated and *Trichuris* positively correlated with bacterial and fungal richness in the gut [100]. These findings highlight the complex and significant interactions between helminths, gut microbiota, and the immune system, pointing to yet another promising research avenue for uncovering new insights into the pathogenesis of asthma.

Asthma is a highly heterogeneous disorder, making it difficult to conduct uniform research and achieve meaningful and generalizable findings [101,102]. Recognizing and characterizing the different inflammatory endophenotypes within asthma would significantly strengthen studies that examine the impact of the gut microbiome on this condition. This approach includes identifying and measuring specific biomarkers of inflammation, such as those associated with type 2 inflammation (e.g., sputum and blood eosinophils, exhaled nitric oxide, serum IgE) as well as non-type-2 inflammation. Doing so not only aids in the precise characterization of asthma but also has potential implications for other airway diseases like chronic obstructive pulmonary disease (COPD) and bronchiectasis, and other atopic diseases which can also involve type 2 inflammation [103,104].

Finally, understanding the impact of various environmental factors, including dietary habits, illnesses, and medications, on the gut mycobiome is essential for developing new therapeutic strategies. Studies have shown that the composition of the gut mycobiome changes according to dietary patterns, with certain diets affecting the prevalence of specific fungi such as *Candida*. For example, one study observed that *Candida* was positively associated with diets high in carbohydrates, but negatively associated with diets rich in amino acids, proteins, and fatty acids [105]. Additionally, therapeutic strategies depend on expanding in vitro and animal research to include the specific mechanisms of gut–lung crosstalk discussed in this review, such as syk-dependent pathways in the gut and the role of ILC2 cells in the lungs.

## 6. Conclusions

This review highlighted the significant yet complex role of the gut mycobiome in asthma, underscoring the intricate interplay between intestinal fungi and respiratory health. Through the advent of metagenomic sequencing, we have been able to probe the character of the gut mycobiome in asthma patients, uncovering significant differences in fungal composition. Mechanisms linking the gut mycobiome and asthma, such as fungal sensing by host cells and fungal-derived metabolites, were explored, shedding light on how intestinal dysbiosis can influence and exacerbate asthma. Strategies for modulating the gut mycobiome, whether implemented in isolation or combined, hold promise in the prevention and management of asthma.

Further research is needed to fully understand these complex interactions and their therapeutic potential. This includes better characterizing the inflammatory endophenotypes of asthma to reduce heterogeneity in research findings and examining the influence of environmental factors, such as diet, on the gut mycobiome. Moreover, delving into specific pathways, such as Syk-dependent mechanisms and the role of ILC2 cells in the gut–lung crosstalk enhancing asthma, presents promising avenues for novel treatments. By continuing to unravel the complexities of the gut mycobiome and its connections to respiratory diseases, we can move closer to more personalized and effective management strategies for asthma, potentially extending these insights to other related airway diseases.

## Figures and Tables

**Figure 1 jof-10-00192-f001:**
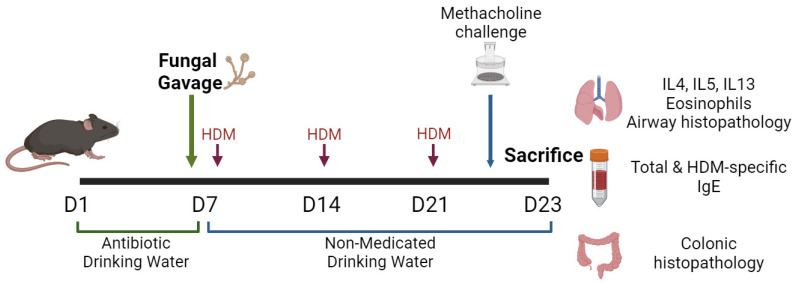
A murine model of gut fungal dysbiosis and allergic airway inflammation. First, mice are treated with antibiotic drinking water (e.g., Cefoperazone 0.5mg/mL) for 7 days to deplete intestinal bacteria. Live fungi (e.g., 10^7^ *Candida albicans* yeast cells) are then administered in high concentrations through gavage feeding. Intranasal house dust mite (HDM) sensitization is performed once weekly for a total of three times. Stool samples are analyzed at different time points to confirm fungal colonization. The study ensures no colonic inflammation by histopathology, or fungal presence in the lungs beyond control baseline (e.g., by real-time polymerase chain reaction). It is also important to conduct multiple repetitions of the experiment, use enough mice in each, and ensure that only the specific intervention (i.e., fungal gavage) differentiates the control and experimental groups. Created with BioRender.com, accessed on 1 February 2024.

**Figure 2 jof-10-00192-f002:**
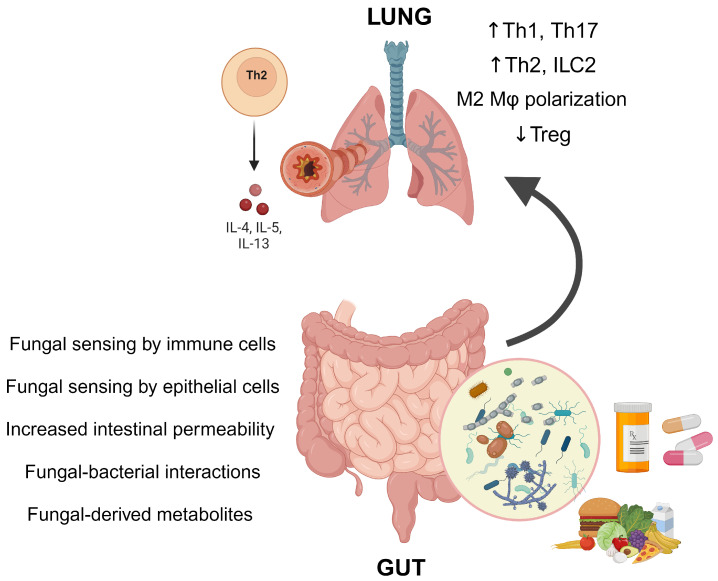
Major mechanisms identified in studies investigating the association between intestinal fungal dysbiosis and asthma, focusing on how fungal dysbiosis exerts its effects in the gut. Interactions between fungi, host cells, and other intestinal microbes may, through complex crosstalk, enhance airway inflammation. This is characterized by the upregulation of Th1 and Th17 immune responses, the promotion of type 2 inflammation (i.e., Th2 and ILC2), and the downregulation of Tregs. Tregs are essential for modulating immune responses and play a pivotal role in controlling asthma. ILC2: group 2 innate lymphoid cells; Mφ: macrophage; Treg: regulatory T cell. Created with BioRender.com, accessed on 1 February 2024.

**Table 1 jof-10-00192-t001:** Overview of research on gut mycobiome in patients with asthma.

Study	Region	Design	Methods	Outcome	Gut Mycobiome Associated with Outcome
Pediatrics					
Fujimura et al., 2016 [15]	Detroit, MI, United States	Cohort (*n* = 298 with available stool samples)	Stool sample collected at age 1–11 months. ITS2 gene amplifications	Parental report of physician-diagnosed asthma at 4 years	Increased: *Candida*, *Rhodotorula*, *Debaryomyces*, *Meyerozyma*, *Nigrospora*, *Saccharomyces*, *Pyrenochaetopsis*, *Phanerochaete* Decreased: *Malassezia*
Arrieta et al., 2018 [17]	Rural Ecuador	Case-control (*n* = 27 with atopic wheeze; *n* = 70 healthy controls)	Stool sample collected at age 3 months. 18S gene amplification	Maternally reported atopic wheeze at 5 years	Increased: *Pichia kudriavzevii* Decreased: *Saccharomyces* and Saccharomycetales
Depner et al., 2020 [19]	Rural Austria, Finland, France, Germany, and Switzerland	Cohort (*n* = 930)	Stool sample collected at age 2–12 months. ITS1 gene amplifications	Physician-diagnosed asthma OR parental reports of bronchitis at 6 years	Decreased: *Alternaria* (at 2 months) Bacterial but not fungal maturation is protective for asthma. *Alternaria* at 2 months was associated with bacterial maturation
Schei et al., 2021 [24]	Norway	Secondary data analysis of RCT (*n* = 180)	Stool sample collected at four time points between age 0–24 months. ITS1 gene amplifications	Parental report of physician-diagnosed asthma at 6 years	Increased: Fungal abundance at 2 years
Adults					
Huang et al., 2022 [21]	Shanghai, China	Case-control (*n* = 26 with asthma on ICS; *n* = 12 with asthma not on ICS; *n* = 21 healthy controls)	Stool sample collected at time of interview. ITS gene amplifications	Asthma on ICS vs. healthy controls	Increased: *Russula*, *Sebacina*, *Nectria*, and *Wallemia* in those with asthma on ICS
Kanj et al., 2023 [22]	United States	Cross sectional (*n* = 24 with physician-diagnosed asthma)	Stool sample collected at time of interview. *Candida* gene amplifications	Severe asthma exacerbation in the past year	Increased: *Candida* to bacterial ratio

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
