# Peer review of "Gut Mycobiome and Asthma"

_jof, 2024, doi:10.3390/jof10030192_

Round 1
Reviewer 1 Report
Comments and Suggestions for Authors
In this article by Kanj & Skalski (2023), the authors reviewed aspects related to the intestinal mycobiota and asthma in humans and murine models. The topic is relevant, the text is well written, but some points need to be better discussed and answered before this article is published. Please see below.
Major comments
1) How does this review stand out and differ from those below, published around four years ago?
VAN TILBURG BERNARDES, Erik; GUTIERREZ, Mackenzie W.; ARRIETA, Marie-Claire. The fungal microbiome and asthma. Frontiers in Cellular and Infection Microbiology, v. 10, p. 583418, 2020.
Wardlaw, A. J., Rick, E. M., Pur Ozyigit, L., Scadding, A., Gaillard, E. A., & Pashley, C. H. (2021). New perspectives in the diagnosis and management of allergic fungal airway disease. Journal of Asthma and Allergy, 557-573.
Tiwary, M., & Samarasinghe, A. E. (2021). Initiation and pathogenesis of severe asthma with fungal sensitization. Cells, 10(4), 913.
2) line 103: i) why aren't broad-spectrum antimycotics applied alongside the antibiotics? ii) can't the intrinsic, remaining mycobiota be a noise variable in these in vivo experiments? iii) in addition, some antibiotics can also affect the fungi, which would result in reduced diversity in both the treated and control groups. Discussing these issues will enrich the manuscript and the prospect of future experiments.
3) lines 138-139 and wherever else it appears: it is difficult to find studies dedicated to the pathological effects of specific species as described in this period. I suggest you include the dosage (CFU/g, for example), route of administration and treatment period in the period itself. Or create a table like table 1 to describe these and the other studies mentioned in the manuscript.
4) line 157 and wherever else it appears: i) considering that the action of probiotic microorganisms is strain-specific, please include the code/identification of the strains mentioned throughout the manuscript. ii) I suggest doing this for the other strains, even if pathogenic, this type of traceability will allow a stronger comparison between future studies.
5) Figure 2 is very general, please complete it with more details on the different axis modulation paths.
6) line 179: by fungus-bacteria interaction? ii) And/or by the production of some fatty acids by the intestinal fungi themselves? In this case, I suggest discussing further who they are, what they produce, what their substrate is, etc.
Minor comments
1) line 19: where are the keywords?
2) line 35: please check the formatting.
3) Line 43: please put italics.
4) line 47 and where else it appears: please substitute the term “flora” for microbiota/microbiome.
5) line 66 and where else it appears: please put italics in scientific names.
6) line 137: please check the formatting.
7) line 243: please put italics.
Author Response
In this article by Kanj & Skalski (2023), the authors reviewed aspects related to the intestinal mycobiota and asthma in humans and murine models. The topic is relevant, the text is well written, but some points need to be better discussed and answered before this article is published. Please see below.
R. Thank you for taking the time to review our article. We have made substantial changes to the article based on your constructive feedback, including the addition of a new section entitled "Modulating the Gut Mycobiome in Asthma." Please find below a point-by-point response to your comments.
Major comments
1) How does this review stand out and differ from those below, published around four years ago?
R. We thank the reviewer for pointing out these three narrative reviews. We reviewed all three in details, and have updated our manuscript accordingly. We believe that our current review is both novel and sufficenienctly different from all three and is therefore worth sharing.
VAN TILBURG BERNARDES, Erik; GUTIERREZ, Mackenzie W.; ARRIETA, Marie-Claire. The fungal microbiome and asthma. Frontiers in Cellular and Infection Microbiology, v. 10, p. 583418, 2020.
R. This is a comprehensive review of the mycobiome in asthma, including, in addition to the gut mycobiome, a focus on the lung mycobiome. The review was published in November 2020. At that time, four of the six major human studies that we presented in Table 1 were not available. Also, the review doesn’t delve into important mechanisms, some of which, like the role of ILC2 and C-type lectin receptors in intestinal fungal gut-lung axis interactions, were not well-known back then. This review is, however, important and worth citing, and it also contains a section 'Fungi as Targets for Asthma Treatment and Control'. Therapeutic targeting of the gut mycobiome has been raised by another reviewer, and we felt it is worth discussing further. We have cited this reference and discussed in details the modulation of the gut mycobiome in asthma in our revised article.
Wardlaw, A. J., Rick, E. M., Pur Ozyigit, L., Scadding, A., Gaillard, E. A., & Pashley, C. H. (2021). New perspectives in the diagnosis and management of allergic fungal airway disease. Journal of Asthma and Allergy, 557-573.
R. This article discusses allergic fungal airway disease characterized by airway sensitization to thermotolerant fungi like Aspergillus. It is relevant in highlighting the role of inhaled fungi in the development/progression of lung disease but does not discuss fungi in the gut. We have discussed it in the introduction and cited this article: “However, the healthy gut also contains commensal fungi, some of which, like Aspergillus and Penicillium, are known asthma triggers when inhaled (8, 9).”
Tiwary, M., & Samarasinghe, A. E. (2021). Initiation and pathogenesis of severe asthma with fungal sensitization. Cells, 10(4), 913.
R. This is a very well-written review on the immunologic parameters surrounding fungal allergies with an emphasis on mouse models. The gut mycobiome is only discussed superficially; however, commonalities exist, and we have used this reference to discuss the role of antifungals and fungal elements as potential ways to modulate the mycobiome.
2) line 103: i) why aren't broad-spectrum antimycotics applied alongside the antibiotics? ii) can't the intrinsic, remaining mycobiota be a noise variable in these in vivo experiments? iii) in addition, some antibiotics can also affect the fungi, which would result in reduced diversity in both the treated and control groups. Discussing these issues will enrich the manuscript and the prospect of future experiments.
R. Thank you for raising these important points. Intrinsic mycobiota can definetly be a noise variabe in in-vivo experiments which is why we have recommended in the revised text to"tconduct multiple repetitions of the experiment, use a sufficient number of mice in each, and ensure that only the specific intervention differentiates the control and experimental groups". Also "in cases where establishing colonization is challenging, antifungals may be employed selectively to manage competing fungal species". From our experience, this is usually not needed but we acknowledge there is limited current understanding.
3) lines 138-139 and wherever else it appears: it is difficult to find studies dedicated to the pathological effects of specific species as described in this period. I suggest you include the dosage (CFU/g, for example), route of administration and treatment period in the period itself. Or create a table like table 1 to describe these and the other studies mentioned in the manuscript.
R. We appreciate the reviewer's suggestion to provide more detailed information regarding the dosage, route of administration, and treatment period. In the revised manuscript, we now specify: "Single oral gavage of 5 x 106 live conidia of Aspergillus amstelodami and "Followed by a gavage of 109 CFU/mL of S. cerevisiae..." etc.
4) line 157 and wherever else it appears: i) considering that the action of probiotic microorganisms is strain-specific, please include the code/identification of the strains mentioned throughout the manuscript. ii) I suggest doing this for the other strains, even if pathogenic, this type of traceability will allow a stronger comparison between future studies.
R. We have meticulously revised line 157 and all relevant sections throughout the manuscript to include the specific codes/identifications of the probiotic strains discussed. In the manuscript, we now specify: "Lactobacillus reuteri DSM 17938”, "Bifidobacterium breve B632” and "Lactobacillus salivarius LS01” etc.
5) Figure 2 is very general, please complete it with more details on the different axis modulation paths.
R. We agree with the reviewer's observation regarding Figure 2 and have updated it accordingly.
6) line 179: by fungus-bacteria interaction? ii) And/or by the production of some fatty acids by the intestinal fungi themselves? In this case, I suggest discussing further who they are, what they produce, what their substrate is, etc.
R. The study by Li et al. found that in pigs, the Intestinal fungi Tomentella and Loreleia, have been linked to higher levels of SCFAs. These are not known to produce SCFAs and therefore it is likely by fungus-bacteria interaction as the reviewer correctly points out. The sentence was adjusted in this revised manuscript to reflect this.
Minor comments
1) line 19: where are the keywords?
R. Keywords have been added back to the document: gut-lung axis; asthma; gut mycobiome; fungal dysbiosis; Candida; immune pathways.
2) line 35: please check the formatting.
R. The formatting has been adjusted. Thank you for bringing this to our attention.
3) Line 43: please put italics.
R. We agree with the reviewer's suggestion to use italics for scientific terminology. However, in line with the guidelines from many publishers, including MDPI, we are advised not to italicize Latin words, including terms like "in vitro." For reference: https://blog.mdpi.com/2022/06/09/how-to-use-italics/
4) line 47 and where else it appears: please substitute the term “flora” for microbiota/microbiome.
R. Line 47 and elsewhere: The term “flora” has been replaced with “microbiota/microbiome” throughout the document, specifically in lines 47 and 136, to reflect current scientific terminology accurately. Thank you.
5) line 66 and where else it appears: please put italics in scientific names.
R. Line 66 and elsewhere: we have Italicized all scientific names. Thank you.
6) line 137: please check the formatting.
R. Formatting issues, including the removal of an unintended underline, have been corrected.
7) line 243: please put italics.
R. As per the discussion on italics and following the MDPI guidelines, we have carefully considered the use of italics. Please refer to the MDPI blog link provided above.
Reviewer 2 Report
Comments and Suggestions for Authors
I consider that this manuscript contains a new and important information about the role of fungi in allergic diseases and the important role of the interkindom microorganisms interactions in the gut.
I know that the manuscript is scientifically sound and there are minor corrections that I can suggest the authors , for example :
1. When authors mention the action of probiotic microorganisms is strain-specific, authors need to mention the code of the strains in the entire manuscript, every time authors refer to strains you need to include that information this is useful to other authors in the field.
2. All the scientific names of microorganisms authors mention should be in italics to homogenize the text.
3. When authors mention the bacteria- fungus interaction in the paragraph around line 178-180 authors can add some specific mechanisms of interaction, some fungus products, or other mechanisms that authors can mention.
4. I suggest adding at least a paragraph related with the potential role of intestinal parasites (protozoa and or helminths) in their review in face of recent evidence about the modulator role of allergic responses of some of these organisms as part of the intestinal ecosystem. Is there something interesting to mention about it?
Author Response
I consider that this manuscript contains a new and important information about the role of fungi in allergic diseases and the important role of the interkingdom microorganisms interactions in the gut.
R. We are thankful for your constructive feedback, which has significantly enriched the manuscript.
I know that the manuscript is scientifically sound and there are minor corrections that I can suggest the authors , for example :
1.When authors mention the action of probiotic microorganisms is strain-specific, authors need to mention the code of the strains in the entire manuscript, every time authors refer to strains you need to include that information this is useful to other authors in the field.
R. We are grateful for the emphasis on the importance of specifying the strains of probiotic microorganisms used in studies. Following your advice, we have meticulously updated the manuscript to include the specific strain codes every time probiotic strains are mentioned. Examples now included in the manuscript are Lactobacillus Reuteri DSM 17938, Bifidobacterium breve B632, and Lactobacillus salivarius LS01 etc.
2.All the scientific names of microorganisms authors mention should be in italics to homogenize the text.
R. We appreciate your attention to detail regarding the formatting of scientific names. We have conducted a thorough review of the manuscript and corrected instances where scientific names were not previously italicized
3.When authors mention the bacteria- fungus interaction in the paragraph around line 178-180 authors can add some specific mechanisms of interaction, some fungus products, or other mechanisms that authors can mention.
R. Thank you for your valuable comment. The following paragraph was expanded and modified accordingly: “Another potential communication pathway between gut fungi and the lung involves fungal interactions with commensal bacteria in the gut. Fungi interact with bacteria by direct physical contact, through the secretion of chemical molecules, and by utilizing metabolic by-products and competing for nutrients (33). As previously mentioned, Alternaria can enhance gut bacterial maturation, contributing to protection against childhood asthma (19). Similarly, bacteria can influence fungal maturation and dysbiosis. For example, Enterococcus faecalis prevents fungal overgrowth and balances the microbiome early in life (34). This has been linked to a reduction in Th17 stimulation and a subsequent decrease in the risk of allergic diseases (35). Lactobacillus reuteri is another bacterium that attenuates asthma in mice, doing so through the induction of Treg cells (36). Interestingly, L. reuteri DSM 17938, commonly used in probiotic products, is well known for its antifungal effects, particularly against P. kudriavzevii and other Candida species that are in turn associated with airway inflammation when expanded in the gut (17, 37, 38).”
4. I suggest adding at least a paragraph related with the potential role of intestinal parasites (protozoa and or helminths) in their review in face of recent evidence about the modulator role of allergic responses of some of these organisms as part of the intestinal ecosystem. Is there something interesting to mention about it?
R. Thank you for suggesting the inclusion of a paragraph on the potential modulatory role of intestinal parasites in allergic responses. We have added a new paragraph to Future Directions in Gut Mycobiome and Asthma Research to the manuscript that discusses recent evidence on the role of protozoa and helminths, as well as viruses and archaea, within the intestinal ecosystem.
"In addition to bacteria and fungi, the human gut hosts viruses, archaea, and parasites. The interactions between these organisms and fungi in the gut remain largely unexplored, with very few studies investigating the role of these organism in asthma. One study of the fecal virome in 647 infants found a significant association between several Caudovirales families and increased risk of asthma later in life, independent of bacteria (90). This association was guided by a host's toll-like receptor-9 gene variant, highlighting interactions between phages and the host immune system (90). Another study using fecal samples from 472 children aged 6 to 10 years showed that the archaea Methanosphaera stadtmanae, detected in 78% of samples, was associated with a lower risk of asthma during this age range (91). Regarding helminths, or parasitic worms, their relationship with asthma and allergies is nuanced, with both the potential to exacerbate and mitigate these conditions (92). Research highlights how intestinal helminths can modulate immunity and inflammation through the induction of Treg cells and regulatory macrophages (92). Specifically, a study on Ascaris lumbricoides, Trichuris trichiura, and Toxocara spp. suggested that helminthiasis may enhance Th2 immune responses and foster an immune regulatory phenotype that suppresses allergic reactions, although no direct link to asthma was identified (93). Conversely, a case-control study found that Trichuris trichiura, but not other helminths, in the stools of children aged 6-14 years is associated with asthma, with detection of helminths achieved through flotation and microscopic analysis (94). Although no research has directly examined the interactions between parasites and gut mycobiota in patients with asthma, investigations in non-asthmatics suggest potential interactions, with Strongyloides negatively correlated and Trichuris positively correlated with bacterial and fungal richness in the gut (95). These findings highlight the complex and significant interactions between helminths, gut microbiota, and the immune system, pointing to yet another promising research avenue for uncovering new insights into the pathogenesis of asthma"
Reviewer 3 Report
The topic is very interesting and highly deserving. However, this review paper is poorly organized and also poor in data and information. It seems the authors just read about the Gut microbiome (Mycobiome & Bacteriome) and Asthma through the literature search.
Upon reading it, I was left with the impression that the entire article serves as an introduction, and the true essence of the content is lacking.
The article requires a substantial increase in data and additional chapters.
One question I have is why important references from the last three years were not utilized, such as the following:
van Tilburg Bernardes E, Gutierrez MW and Arrieta M-C (2020) The Fungal Microbiome and Asthma. Front. Cell. Infect. Microbiol. 10:583418. doi: 10.3389/fcimb.2020.583418
Huang, C., Yu, Y., Du, W., Liu, Y., Dai, R., Tang, W., Wang, P., Zhang, C., Shi, G., 2020. Fungal and bacterial microbiome dysbiosis and imbalance of trans-kingdom network in asthma. Clinical and Translational Allergy 10.. https://doi.org/10.1186/s13601-020-00345-8.
Marathe, S.J., Snider, M.A., Flores-Torres, A.S., Dubin, P.J., Samarasinghe, A.E., 2022. Human matters in asthma: Considering the microbiome in pulmonary health. Frontiers in Pharmacology 13.. https://doi.org/10.3389/fphar.2022.1020133.
Another question that arises is what kind of review article it is. Where are the materials and methods, the data collection tools, and the type of statistical analysis?
The authors must summarize the current and future perspectives and provide guidelines on how knowledge of the gut mycobiome can lead to a therapeutic approach to asthma or the mitigation of symptoms, if not relief.
The topic is very interesting and highly deserving. However, this review paper is poorly organized and also poor in data and information. It seems the authors just read about the Gut microbiome (Mycobiome & Bacteriome) and Asthma through the literature search.
Upon reading it, I was left with the impression that the entire article serves as an introduction, and the true essence of the content is lacking.
The article requires a substantial increase in data and additional chapters.
One question I have is why important references from the last three years were not utilized, such as the following:
van Tilburg Bernardes E, Gutierrez MW and Arrieta M-C (2020) The Fungal Microbiome and Asthma. Front. Cell. Infect. Microbiol. 10:583418. doi: 10.3389/fcimb.2020.583418
Huang, C., Yu, Y., Du, W., Liu, Y., Dai, R., Tang, W., Wang, P., Zhang, C., Shi, G., 2020. Fungal and bacterial microbiome dysbiosis and imbalance of trans-kingdom network in asthma. Clinical and Translational Allergy 10.. https://doi.org/10.1186/s13601-020-00345-8.
Marathe, S.J., Snider, M.A., Flores-Torres, A.S., Dubin, P.J., Samarasinghe, A.E., 2022. Human matters in asthma: Considering the microbiome in pulmonary health. Frontiers in Pharmacology 13.. https://doi.org/10.3389/fphar.2022.1020133.
Another question that arises is what kind of review article it is. Where are the materials and methods, the data collection tools, and the type of statistical analysis?
The authors must summarize the current and future perspectives and provide guidelines on how knowledge of the gut mycobiome can lead to a therapeutic approach to asthma or the mitigation of symptoms, if not relief.
Author Response
The topic is very interesting and highly deserving. However, this review paper is poorly organized and also poor in data and information. It seems the authors just read about the Gut microbiome (Mycobiome & Bacteriome) and Asthma through the literature search.
Upon reading it, I was left with the impression that the entire article serves as an introduction, and the true essence of the content is lacking.
The article requires a substantial increase in data and additional chapters.
R. We sincerely appreciate your constructive criticism and acknowledge the concerns you have raised regarding the organization and depth of our review article. We have made extensive changes to address this feedback, including: 1) adding a new section on modulating the gut mycobiome, 2) expanding on experiments presented from the literature, and 3) adding details wherever possible. We have also included relevant studies from our lab on the topic.
One question I have is why important references from the last three years were not utilized, such as the following:
van Tilburg Bernardes E, Gutierrez MW and Arrieta M-C (2020) The Fungal Microbiome and Asthma. Front. Cell. Infect. Microbiol. 10:583418. doi: 10.3389/fcimb.2020.583418
Huang, C., Yu, Y., Du, W., Liu, Y., Dai, R., Tang, W., Wang, P., Zhang, C., Shi, G., 2020. Fungal and bacterial microbiome dysbiosis and imbalance of trans-kingdom network in asthma. Clinical and Translational Allergy 10.. https://doi.org/10.1186/s13601-020-00345-8.
Marathe, S.J., Snider, M.A., Flores-Torres, A.S., Dubin, P.J., Samarasinghe, A.E., 2022. Human matters in asthma: Considering the microbiome in pulmonary health. Frontiers in Pharmacology 13.. https://doi.org/10.3389/fphar.2022.1020133.
R. Thank you for the suggestion. We acknowledge the importance of the references you have highlighted and appreciate your inquiry regarding their utilization in our manuscript. We have carefully integrated these references throughout our manuscript, citing them where appropriate to support our discussion.
Another question that arises is what kind of review article it is. Where are the materials and methods, the data collection tools, and the type of statistical analysis?
R. We appreciate the question regarding the nature of our review article. It's important to clarify that this manuscript does not constitute a systematic review. Instead, it is a narrative review designed to synthesize and discuss the existing literature on the role of the gut mycobiome in asthma. As such, and like the three other articles by van Tilburg Bernardes et al., Huang et al., and Marathe et al., cited in the previous comment, our review does not follow the structured approach typical of systematic reviews, which would include detailed materials and methods, data collection tools, and statistical analysis.
The authors must summarize the current and future perspectives and provide guidelines on how knowledge of the gut mycobiome can lead to a therapeutic approach to asthma or the mitigation of symptoms, if not relief.
R. Based on your suggestion, we have included a new section of around 1000 words in the manuscript titled “Modulating the Gut Mycobiome in Asthma” where we explore the emerging research and potential strategies for influencing the gut mycobiome to manage asthma, and suggest that a combination of strategies may be needed to effectively and therapeutically alter the gut mycobiome in these patients.
Reviewer 4 Report
The manuscript “Gut Mycobiome and Asthma” reviews how changes in intestinal fungal composition can alter the immune function of the lungs in patients with asthma since several studies indicate that intestinal fungal dysbiosis can elevate the risk of asthma in children and exacerbate it in adults. Furthermore, it is an interesting work, considering that asthma affects an increasing number of people around the world, and the role of intestinal and lung microbiomes seems to have a very important implication since they can drive and aggravate asthma. However, I have some comments that I list below.
Comments
It is mentioned that Candida is one of the fungi most closely related to the intestine-lung interaction; the authors particularly mention C. krusei. Given its greater prevalence in fungal diseases, is it known if C. albicans is also related? Has it been mentioned? What is the species or species that can increase the risk of asthma?
It would be important to mention what the fungal microbiome is in patients who do not have asthma.
Conclusions
I suggest you discuss the importance of the methods used in identifying fungi in the mycobiota since the probable appearance of cryptic species could provide more information on which species are most closely related to asthma patients.
Minor comments
Line 35: Change “(6, 7). However…” to “(6, 7). However…”
Line 35: Change “Candida krusei” to “Candida krusei”
Line 137: Delete underlined “Epicoccum nigrum and”
Line 138: Change “Wallemia mellicola” to “W. mellicola”
I have no comments
Author Response
The manuscript “Gut Mycobiome and Asthma” reviews how changes in intestinal fungal composition can alter the immune function of the lungs in patients with asthma since several studies indicate that intestinal fungal dysbiosis can elevate the risk of asthma in children and exacerbate it in adults. Furthermore, it is an interesting work, considering that asthma affects an increasing number of people around the world, and the role of intestinal and lung microbiomes seems to have a very important implication since they can drive and aggravate asthma. However, I have some comments that I list below.
Comments
It is mentioned that Candida is one of the fungi most closely related to the intestine-lung interaction; the authors particularly mention C. krusei. Given its greater prevalence in fungal diseases, is it known if C. albicans is also related? Has it been mentioned? What is the species or species that can increase the risk of asthma?
R. Thank you for this important question. Our lab works with mouse models of Candida albicans-induced intestinal dysbiosis, which have consistently been shown to enhance sensitization to house dust mites. We apologize for the initial oversight in referring to this simply as Candida without specifying "albicans". This has been addressed in the revised manuscript: “In a mouse model of asthma with Candida albicans-induced gut dysbiosis, an increase in lung-resident ILC2 cells was observed even in the resting state, prior to allergen sensitization.” To our knowledge, the role of C. albicans gut dysbiosis has not yet been demonstrated in humans with asthma (Table 1).
It would be important to mention what the fungal microbiome is in patients who do not have asthma
R. Thank you for your valuable feedback. In response to your suggestion, we have updated the section "Characterizing the Gut Mycobiome in Asthma" to include information about the fungal microbiome in patients who do not have asthma. We have added the following sentence to provide a clear comparison and context: ”Candida, Saccharomyces, and Cladosporium are the predominant constituants of the healthy adult gut mycobiome, with an increase in Candida spp. linked to diseases associated with urbanization and inversely related to bacterial microbiome diversity (zhang et al., 2022)”.
Conclusions
I suggest you discuss the importance of the methods used in identifying fungi in the mycobiota since the probable appearance of cryptic species could provide more information on which species are most closely related to asthma patients.
R. Thank you for this suggestion We acknowledge the critical role of advanced metagenomic sequencing technologies in our exploration of the gut mycobiome in asthma patients. This innovative approach has enabled us to uncover significant differences in fungal composition, including the identification of cryptic species that were previously undetected.We have modified th econclusion of the text to reflect this.
Minor comments
Line 35: Change “(6, 7). However…” to “(6, 7). However…”
Line 35: Change “Candida krusei” to “Candida krusei”
Line 137: Delete underlined “Epicoccum nigrum and”
Line 138: Change “Wallemia mellicola” to “W. mellicola”
R. Thank you for these excellent observations; all have been addressed accordingly.
Round 2
Reviewer 1 Report
One point still needs to be clarified before this manuscript can be published.
Regarding the murine model of fungal colonization, or FMT with a focus on mycobiota, the answer given to this question was very general.
Since current knowledge is limited, I suggest that the authors describe in detail what is known, i) indicate the "gold references" of these techniques, ii) contrast the advantages and disadvantages, limitations, etc., iii) mention which protocols best meet current needs, iv) mention which antibiotics and antimycotics are used, etc. This will enrich the work substantially.
The minor comments were answered satisfactorily.
Author Response
Dear Reviewer,
Thank you for your valuable feedback. We have addressed your questions, detailing different models used for studying intestinal fungal dysbiosis in asthma, highlighting that no single model is universally best, as the approach depends on the objectives of the experiments. We discussed various approaches, including the use of different antibiotics, as well as the advantages and disadvantages of using Specific Pathogen-Free (SPF) or Germ-Free (GF) mice. We have also covered antibiotics and antifungals in details and provided guidance on when to use them. We have added the text below, made changes to the rest of the document, and implemented minor adjustments under Figure 2 for further clarity.
“Developing animal models to investigate the role of intestinal fungal dysbiosis in asthma requires a tailored approach, dependent on the objectives of the experiment. Specific Pathogen-Free (SPF) mice are frequently used for this purpose (23). These mice are free from specified pathogens but not all microorganisms. To promote intestinal fungal colonization, SPF mice are first treated with antibiotics to deplete gut bacterial populations, followed by fungal gavage to establish intestinal fungal colonization (23, 24). In a study investigating the effects of Candida albicans-induced gut dysbiosis on colitis, antibiotics were not utilized. As a result, high doses of C. albicans had to be repeatedly administered via gavage to ensure effective colonization (25). In experiments where mice were premedicated with antibiotics, a single gavage of 107 live C. albicans yeast was sufficient to establish intestinal colonization that persisted for weeks after gavage (22).
In fact, antibiotic treatment to deplete intestinal bacteria in SPF mice is an affordable and widely accessible method. Cefoperazone, a broad-spectrum third-generation cephalosporin, is often selected for its minimal systemic absorption, making it ideal for experiments on asthma where altering the lung bacterial composition is undesirable (22, 23, 26). Cefoperazone sodium salt may be dissolved at a concentration of 0.5 mg/mL in deionized water and provided to mice as the only source of drinking water for 7 days. Other antibiotics that have been utilized, often in combination, include ampicillin (1 mg/mL), clindamycin (0.5 mg/mL), metronidazole (1 mg/mL), streptomycin (5 mg/mL), ciprofloxacin (1 mg/mL), and vancomycin (1 mg/mL) (27-29).
Oral gavage of antibiotics has been employed in murine experiments to ensure the precise administration of antibiotic doses (30). Although daily gavage can be labor-intensive, it may minimize variations in antibiotic intake, both within and among experimental groups, enhancing the likelihood of achieving consistent intestinal colonization. In cases where fungal colonization remains challenging, antifungals may be employed selectively to manage competing fungal species. The antifungal fluconazole has been used in studies aiming to promote the growth of fungi other than Candida (31). In a study of the effect of intestinal C. albicans proliferation on bleomycin-induced pulmonary fibrosis in mice, fluconazole (0.5 mg/ml) was administered to the control group to ensure that no endogenous Candida would remain and proliferate in the gut following bacterial depletion by antibiotics (28). A separate set of experiments was necessary to exclude the possibility fluconazole itself influenced pulmonary fibrosis (28). Amphotericin-B at a concentration of 0.1 mg/ml, has been shown to significantly disrupt fungal populations within the gut. Unlike fluconazole, Amphotericin-B is not absorbed through the gastrointestinal tract (31).
Germ-free mice, maintained in germ-free conditions, offer a valuable alternative to SPF mice for studies requiring colonization with specific, defined organisms (29). These mice are completely devoid of all microorganisms, enabling researchers to precisely control the microbial environment and study the effects of individual intestinal fungi or defined intestinal fungal communities on host physiology and disease. However, these mice can be costly, require specialized equipment and training, and their lack of a native microbiome may limit the generalizability of findings to microbiota-harboring organisms. This limitation is particularly relevant when considering the complex interactions within microbial ecosystems outside the gut, which are lacking in germ-free models (29).”
Reviewer 3 Report
Lines 82, 167: “Candida spp.” Please use italics only for the names of the genus or species but not for spp.
L 108 “Candida albicans-induced gut dysbiosis on colitis”. We only use italics when referring to genera or species isolates, not to a compound word containing a genus or species name.
269 “fungal probiotics”. Please don't mess around with the definition of probiotics. There is no such thing as fungal probiotics; however, there are probiotics that consist of fungal species.
Figure 1: Please enhance the text around Figure 1 by providing more details and placing emphasis on the gold standard technique for investigating the specific topic.
Figure 2: This image contains a significant amount of information. Would it be possible to receive a more detailed description of its contents?
Lines 82, 167: “Candida spp.” Please use italics only for the names of the genus or species but not for spp.
L 108 “Candida albicans-induced gut dysbiosis on colitis”. We only use italics when referring to genera or species isolates, not to a compound word containing a genus or species name.
269 “fungal probiotics”. Please don't mess around with the definition of probiotics. There is no such thing as fungal probiotics; however, there are probiotics that consist of fungal species.
Figure 1: Please enhance the text around Figure 1 by providing more details and placing emphasis on the gold standard technique for investigating the specific topic.
Figure 2: This image contains a significant amount of information. Would it be possible to receive a more detailed description of its contents?
Author Response
Thank you for your continued feedback.
-We have de-italicized 'spp.' across five different instances in the manuscript.
-We have changed “Candida albicans-induced gut dysbiosis in colitis” to "Candida albicans gut dysbiosis in colitis"
-We agree that the term "fungal probiotics" may not align with the established understanding of probiotics and have changed this to “probiotics containing fungal species”
-We have addressed the comments for both figures by expanding the descriptions to encapsulate the complexity of each and to enhance clarity.
Round 3
Reviewer 1 Report
All my questions were answered satisfactorily.
All my questions were answered satisfactorily.
Author Response
Thank you for your insightful feedback throughout the review process